# Cytogenetic Profile of Chromosomal Aberrations in Leukemia Using the Fluorescence In Situ Hybridization (FISH) Method at a Tertiary Institution in Gauteng Province

**DOI:** 10.3390/diagnostics15192429

**Published:** 2025-09-24

**Authors:** Zamathombeni Duma, Karabo C. Matsepane, Koketso Nkoana, Sara M. Pheeha, Bathabile Mbele, Tandekile Simela-Tshabalala, Donald M. Tanyanyiwa

**Affiliations:** 1Department of Chemical Pathology, Sefako Makgatho Health Sciences University, Molotlegi Street, Ga-Rankuwa Zone 1, Pretoria 0208, South Africa; karabomatsepane16@gmail.com (K.C.M.); koketso.nkoana@smu.ac.za (K.N.); pheeha.sm@gmail.com (S.M.P.); bathabilembele@yahoo.com (B.M.); tandekile.simela-tshabalala@smu.ac.za (T.S.-T.); donaldmoshen@gmail.com (D.M.T.); 2National Health Laboratory Service, Dr George Mukhari Academic Hospital, Pretoria 0208, South Africa; 3Faculty of Science, School of Chemistry, University of the Witwatersrand, Johannesburg 2050, South Africa; 4Department of Laboratory Diagnostic and Investigative Sciences, University of Zimbabwe, Mazowe Street, Harare 167, Zimbabwe

**Keywords:** leukemia subtypes, chromosomal aberrations, fluorescence in situ hybridization, conventional karyotyping

## Abstract

**Background:** Leukemia, a hematologic malignancy, is the major fluid tumor. However, there is a paucity in laboratory characterization in South Africa due to limited diagnostic infrastructure. Chromosomal aberrations play a crucial role in leukemia pathogenesis, influencing classification, prognosis, and treatment. **Aim:** This study aimed to characterize chromosomal aberrations in leukemia patients using the fluorescence in situ hybridization (FISH) method, with the goal of improving diagnostic precision and guiding tailored treatment in resource-limited settings. **Methodology:** This study was a retrospective analysis of 349 leukemia patient records from the NHLS Corporate Data Warehouse, covering cases diagnosed between January 2019 and January 2024. Chromosomal aberrations were assessed using FISH, including cases of CML, AML, CLL, and ALL. **Results:** CML was the most prevalent leukemia subtype (40%), followed by AML (31%). Age-specific distributions were significant across subtypes (*p* < 0.0001). FISH detected subtype-specific aberrations: t(1;19) and t(12;21) in 25% of ALL cases; t(8;21) and t(15;17) in 22–33% of AML cases; and t(9;22) in 100% of CML cases. In CLL, 13q deletions were most common (53% complex, 33% simple). **Conclusions:** This study reveals distinct chromosomal aberration patterns in leukemia patients in Gauteng, with CML as the most prevalent subtype. Distinct patterns were observed across ALL, AML, and CLL, with age and gender-specific trends. Findings highlight regional genetic influences, diagnostic gaps, and healthcare challenges, emphasizing the urgent need to expand cytogenetic and molecular testing to enable targeted diagnostics, risk stratification, and personalized therapies in sub-Saharan Africa.

## 1. Introduction

Leukemia represents a significant global public health challenge, contributing substantially to morbidity and mortality [1]. In 2020, 474,519 new cases were reported globally [2]. However, in Africa, the burden of leukemia remains underrecognized, largely due to limitations in oncology infrastructure and diagnostic capacity. For instance, only 23,928 new cases were reported across the continent in 2012 [3], and South Africa’s National Cancer Registry recorded just 568 new cases in 2017 [4]. These figures reflect underdiagnosis and underreporting rather than true disease prevalence [5]. The challenge is further exacerbated by the high burden of infectious diseases such as HIV and tuberculosis, socioeconomic disparities, and constrained healthcare resources, all of which have strained South Africa’s health system and diverted attention from non-communicable diseases such as leukemia [6,7,8].

This current study focuses on the four acute and chronic hematological tumors: acute lymphoblastic leukemia (ALL), acute myeloid leukemia (AML), chronic lymphocytic leukemia (CLL), and chronic myeloid leukemia (CML) [9,10]. These subtypes are often associated with distinct chromosomal aberrations, which have important implications for disease progression, treatment response, and prognosis [11]. Given South Africa’s genetic diversity, investigating the cytogenetic landscape of leukemia within this population could uncover novel or population-specific chromosomal patterns. However, there is a significant gap in cytogenetic data from African populations due to resource limitations and underrepresentation in genomic studies [12].

This study aimed to characterize the cytogenetic profile of chromosomal aberrations in leukemia patients using the fluorescence in situ hybridization (FISH) method, and to describe associated patient demographics. Gaining deeper insights into the molecular characteristics of leukemia patients may improve diagnostic precision and enable the development of personalized treatment approaches, ultimately benefiting South Africa and other resource-limited countries with similar healthcare challenges.

## 2. Materials and Methods

### 2.1. Study Design

This was a retrospective study conducted on a limited number of cases using archived data over a period of 5 years of leukemia patients treated at a tertiary institution in Gauteng province. The nature of the study allowed for the evaluation of chromosomal aberrations in leukemia patients without direct contact or intervention with participants, ensuring minimal risk while enabling the efficient analysis of existing data.

### 2.2. Data Collection

A total of 349 electronic medical records of leukemia patients were retrieved from the National Health Laboratory Service (NHLS) Central Data Warehouse (CDW). Patients were selected for the study based on their chromosomal analysis results obtained through FISH testing. The records, spanning from 1 January 2019 to 1 January 2024, were sourced from a tertiary hospital in Gauteng Province and included patients who had undergone cytogenetic testing. The dataset comprises leukemia patients of all ages and racial backgrounds, diagnosed with chronic myeloid leukemia (CML), acute myeloid leukemia (AML), chronic lymphocytic leukemia (CLL), or acute lymphoblastic leukemia (ALL). For acute cases, only primary acute leukemias were included, while secondary leukemias were excluded.

### 2.3. Sample Preparation Procedure and Laboratory Analysis

#### Fluorescence In Situ Hybridization (FISH) Procedure

Bone marrow aspirates from the first or second draw were preferred for FISH analysis due to superior cellular quality and were collected in lithium heparin vacutainers. Peripheral blood was used when bone marrow was unavailable. The FISH probe panel for detecting chromosomal aberrations is listed in Table 1. Slides were examined using a fluorescence microscope with appropriate filters. At least 200 interphase nuclei were analyzed per case, and signal patterns were interpreted per established guidelines. Cutoff values varied depending on the probe, target gene or chromosome, and clinical context. For a positive diagnosis, a cutoff of ≤5% was used. Results were reported according to the International System for Human Cytogenomic Nomenclature (ISCN) guidelines. Senior staff confirmed the results of chromosomal aberrations.

### 2.4. Data Analysis

Patient data were recorded in Microsoft Excel and analyzed using GraphPad Prism version 5 (GraphPad Software, Inc., San Diego, CA, USA). All data were analyzed using the Chi-squared (χ^2^) test to compare demographic and clinical characteristics, such as age and gender, in relation to patterns of chromosomal aberrations across acute and chronic leukemia subtypes in the study population. Additionally, age-related demographic differences across leukemia subtypes, as presented in Table 2, were assessed using one-way analysis of variance (ANOVA). Descriptive statistics, including demographic and clinical features, were reported as frequencies (*n*) and percentages (%). A *p*-value of less than 0.05 (*p* < 0.05) was considered statistically significant.

## 3. Results

### 3.1. Distribution of Demographic Characteristics and Leukemia Subtypes Identified by FISH

The study population demographic characteristics are summarized in Table 2. The study involved 349 patients diagnosed with various types of leukemia, categorized into four subtypes: Acute lymphoblastic leukemia (ALL), Acute myeloid leukemia (AML), Chronic lymphocytic leukemia (CLL), and Chronic myeloid leukemia (CML). The median age of the total study population was 32 years (range: 16–51 years), with significant differences in leukemia subtypes observed across different age groups (*p* < 0.001). There were 173 male (49.6%) and 176 female (50.4%) patients, showing almost equal gender distribution. Notably, the study population consisted entirely of Black patients (*n* = 349, 100%). Among the four leukemia subtypes, CML was the most prevalent, accounting for 40% (*n* = 139) of cases, followed by AML at 31% (*n* = 109). ALL represented 19% (*n* = 65), while CLL was the least common subtype, comprising only 10% (*n* = 36) (Table 2).

**Table 2 diagnostics-15-02429-t002:** Demographic profile of individuals diagnosed with leukemia in the study population.

Demographic Characteristics	Total Population (*n* = 349)	ALL (*n* = 65)	AML (*n* = 109)	CLL (*n* = 36)	CML (*n* = 139)	*p*-Value
Age: Median (interquartile range)	32 (16–51)	12 (6–21)	22 (13–40)	68 (56–75)	39 (29–46)	*p* < 0.001
Gender:						*p* = 0.1440
Male, *n* (%)	173 (50%)	28 (16%)	48 (28%)	21 (12%)	76 (44%)	
Female, *n* (%)	176 (50%)	37 (21%)	61 (34.5%)	15 (8.5%)	63 (36%)	
Ethnicity:						
Black, *n* (%)	349 (100%)	65 (19%)	109 (31%)	36 (10%)	139 (40%)	

### 3.2. Age Distribution Patterns Across Four Main Leukemia Subtypes

The findings of this study demonstrate that leukemia subtypes exhibit distinct age-related prevalence patterns. The ALL was most prevalent in children aged 1–10 years, accounting for 48% (31 of 65) of cases within this group. Interestingly, two children with Down syndrome were identified among the ALL cases. AML cases were more common among young adults aged 21–40 years, representing 33.3% (36 of 109) of AML cases. In contrast, CML was most prevalent in the 41–60-year age group (44%, 61 of 139 cases), and while CLL was most frequently diagnosed in individuals aged 61–70 years (36%, 13 of 36 cases) and those over 70 years (33%, 12 of 36 cases). These differences in age distribution among leukemia subtypes were statistically significant (*p* < 0.001), as presented in Table 3 and Figure 1.

### 3.3. Gender Distribution and Incidence Patterns of Leukemia Subtypes

In this study population (*n* = 349), there were 173 male and 176 female leukemia patients with a fairly even gender distribution among the four main leukemia subtypes. A similar trend was observed among AML and ALL patients in this study population, with females showing a slightly higher incidence than males. The overall proportion of females was 34.5% (61 out of 176) in AML cases and 21% (37 out of 176) in ALL cases. In contrast, CLL and CML cases were more prevalent in males than females, with males accounting for 44% (76 out of 173) of CML cases and 12% (21 out of 173) of CLL cases. No statistically significant differences were observed between males and females across the four main leukemia subtypes (*p* = 0.1440) (Figure 2).

### 3.4. Prevalence of Common Chromosomal Aberrations Detected by FISH in Leukemia Patients

The analysis in this section is limited to the most commonly observed chromosomal aberrations in the study population, excluding less frequent findings. Among patients diagnosed with ALL (*n* = 65), the most frequently observed chromosomal aberrations were t(1;19), associated with the *TCF3::PBX1* fusion probe, and t(12;21), linked to the *ETV6::RUNX1* fusion probe, each identified in 16 cases (25%).

In AML (*n* = 109), the most prevalent chromosomal aberration was the t(8;21) translocation, involving the *RUNX1::RUNX1T1* probe, detected in 35 cases (33%). This was followed by the t(15;17) translocation, associated with the *PML::RARA* fusion probe, found in 29 AML cases (22%). A noteworthy finding across both ALL and AML cases was the t(9;22) translocation, associated with the *BCR::ABL1* fusion. This aberration was detected in 26 patients, comprising 14 ALL cases (22%) and 12 AML cases (11%). Additionally, the *11q23* aberration associated with the *KMT2A* probe was detected in 6 ALL (9%) and 14 AML (13%) cases.

In the chronic leukemia group, analysis showed that among patients with CLL (*n* = 36), the most common chromosomal aberration was a complex rearrangement involving *13q34/13q14.3* and *17p13.1/11q22.3*, associated with *CEP12* and *TP53::ATM* probes, which was detected in 19 cases (53%). This was followed by a simple deletion at *13q34/13q14.3*, detected in 12 cases (33%) using the CEP12 probe. In CML (*n* = 139), the t(9;22) translocation associated with the *BCR::ABL1* probe was detected in all 139 cases (100%) (Table 4).

### 3.5. Gender-Specific Distribution and Prevalence of Chromosomal Aberrations Across Leukemia Subtypes

This study examined gender-specific distribution patterns of the most frequently observed chromosomal aberrations. In ALL cases (*n* = 65), the chromosomal aberrations *11q23*, t(12;21), and t(9;22) were more frequently detected in females, with prevalence rates of 8% (*n* = 5), 20% (*n* = 13), and 12% (*n* = 8), respectively. In contrast, the t(1;19) translocation was more prevalent in males (14%, *n* = 9). A significant difference in the distribution of these aberrations among ALL cases was noted between males and females (*p* < 0.05) (Figure 3A). Among AML cases (*n* = 109), chromosomal aberrations t(9;22), *11q23*, t(8;21), and t(15;17) were mostly observed in females at the frequencies of 7% (*n* = 8), 8% (*n* = 9), 18% (*n* = 20), and 14% (*n* = 15), respectively. Furthermore, no significant difference was observed between males and females in these cases (*p* = 0.896) (Figure 3B).

In CLL cases (*n* = 36), *13q34/13q14.3* and the combination of *13q34/13q14.3* with *17p13.1/11q22.3* were more common in males, with prevalence rates of 22% (*n* = 8) and 31% (*n* = 11), respectively (Figure 3C). Interestingly, no significant difference was observed between males and females across these aberrations (*p* = 0.500). Similarly, in CML cases (*n* = 139), the t(9;22) translocation was primarily observed in males, accounting for 55% (*n* = 76) of cases. However, no significant difference was observed between males and females (*p* = 0.517) (Figure 3D). Notably, the t(9;22) chromosomal aberration showed contrasting gender patterns across leukemia subtypes, being more prevalent in females with ALL and AML, whereas it was more common in males with CML (Figure 3).

### 3.6. Distribution of Chromosomal Aberrations Across Age Groups in Leukemia Patients

Age-related patterns of chromosomal aberrations in lymphocytic and myelogenous leukemia patients were evaluated. In acute lymphocytic leukemia (ALL), distinct age-specific distributions were observed: t(1;19) and t(12;21) were predominant in children under 12 years (18%, (12/65) and 17%, (11/65), respectively), while t(9;22) was more frequent in individuals aged 13–39 years at 14% (9/65). The *11q23* aberration was detected in age groups under 12 years and 13–39 years at 5% (3/65). Despite these trends, no statistically significant difference was found between age and chromosomal aberrations in ALL cases (*p* = 0.0855, Figure 4A). In contrast, CLL showed chromosomal abnormalities primarily in adults over 60 years, with *13q34/13q14.3*; *17p13.1/11q22.3* at 39% (14/36) and *13q34/13q14.3* at 22% (8/36) being the most common. However, similar to ALL, there was no statistically significant correlation between age and chromosomal aberrations in CLL patients (*p* = 0.4643, Figure 4B).

Among patients with myelogenous leukemia, the most frequently observed chromosomal aberration in AML was t(15;17), detected in 18% (20/109) of cases, primarily affecting individuals aged 13–39 years. This was followed by t(8;21), present in 17% (19/109) of cases, and most prevalent in patients under 12 years. The t(9;22) aberration was identified in 5% (5/109) of AML cases, occurring mainly in the 13–39 and 40–59 age groups. Similarly, *11q23* aberrations were found in 5% (5/109) of cases, predominantly among patients younger than 12 years and those aged 13–39. In contrast to ALL cases, a statistically significant difference between chromosomal aberrations and age group was observed in AML (*p* = 0.0489; Figure 4C). In CML cases, t(9;22) was the sole chromosomal aberration detected. It was most common in patients aged 13–39 years (44%; 61/139), followed by those aged 40–59 years (36%; 50/139), and individuals over 60 years (17%; 24/139). No significant difference in the frequency of chromosomal aberrations was observed across age groups in CML (*p* = 0.6162, Figure 4D).

## 4. Discussion

This study aimed to address the critical gap in data on the cytogenetic profile of leukemia in African populations by identifying and characterizing chromosomal aberrations across different leukemia subtypes, using the FISH method, and examining their associations with patient demographics. Most existing leukemia research is derived from Western cohorts, leaving African genetic diversity largely understudied [13]. This scarcity of region-specific studies has left a gap in local genetic data essential for developing targeted diagnostic and treatment guidelines for African patients, many of whom continue to face restricted access to adequate leukemia care across much of Sub-Saharan Africa [14].

The results provided crucial insights into the prevalence and distribution of chromosomal aberrations in this study population of leukemia patients. It is worth noting that 40% of the patients had CML, showing its heavy burden in the cohort. The high prevalence of CML aligns with findings from other studies in Sub-Saharan Africa, such as in Eritrea, where CML was the most common subtype with a prevalence of 35.4% [15]. In support of this, Korubo et al., in Nigeria, also noted CML as the predominant leukemia subtype, reaffirming its significant burden in the African population [16]. The consistency of these findings across different regions suggests that CML may represent a significant proportion of leukemia cases in African populations, potentially influenced by a combination of genetic predisposition, environmental exposures, and healthcare access factors [17,18].

This study further examined the most prevalent chromosomal abnormalities detected across various leukemia subtypes, identifying key patterns that align with previous findings. In ALL cases, t(1;19) and t(12;21) were the most frequently observed aberrations, while AML cases were predominantly characterized by t(8;21) and t(15;17) aberrations. The notably higher occurrence of t(15;17) and t(8;21) observed in this cohort, compared to Western populations, may reflect the unique genetic diversity of African populations, as well as environmental influences such as exposure to infectious agents, chemicals, and radiation, and the impact of late diagnosis [19]. The detection of t(9;22) and *11q23* rearrangements in both ALL and AML highlights their crucial role in acute leukemias. According to the literature, 11q23 rearrangements cause oncogenic fusions of the *KMT2A::MLL* gene, leading to disruption of normal blood cell development. Therefore, these findings emphasize the importance of chromosomal profiling for diagnosis, risk stratification, and treatment guidance in acute leukemia. In CLL patients, the most common detected chromosomal aberration was a complex aberration involving regions *13q34/13q14.3* and *17p13.1/11q22.3*, followed by isolated deletions of *13q34/13q14.3*. These findings are consistent with previous studies identifying 13q deletions as the predominant cytogenetic change in CLL [20,21,22]. Importantly, to note in CML patients, the hallmark t(9;22) translocation was the predominant aberration, consistent with its near-universal presence in CML cases [23,24].

As illustrated in Figure 2 and Figure 3, this study revealed distinct gender-based patterns in both the distribution of leukemia subtypes and associated chromosomal aberrations. In acute leukemias (AML and ALL), a higher proportion of cases occurred in females, AML (34.5%) and ALL (21%) which may be attributed to several biological and behavioral factors including hormonal influences, particularly oestrogen, are known to play a role in hematopoiesis and leukemogenesis which may influence leukemia susceptibility in females [25]. Additionally, women may be more likely to seek medical attention earlier than men, resulting in a higher reported prevalence of leukemia cases among females [26]. In ALL and AML, females showed a higher frequency of aberrations, including 11q23, t(12;21), t(9;22), t(8;21), and t(15;17), potentially driven by both hormonal and genetic factors. Oestrogen, for example, interacts with *KMT2A::MLL1* at *11q23* and may trigger DNA double-strand breaks through oestrogen receptor signaling, increasing the risk of *11q23* aberration in females [27]. Furthermore, some X-linked genes can escape inactivation in females, resulting in increased expression of certain genes that could influence the development and progression of leukemia [28]. In contrast to acute leukemias, chronic leukemia cases in the present study demonstrated a different trend, with CML (44%) and CLL (12%) being more prevalent in males. This male predominance may be explained by several factors, including greater exposure to occupational carcinogens and benzene in male-dominated industries, as well as lifestyle factors such as smoking, which are more prevalent among men and contribute to an increased risk of leukemia [29,30,31,32]. Cytogenetic aberrations such as *13q34/13q14.3* and *17p13.1/11q22.3* in CLL, and the hallmark t(9;22) translocation in CML, were observed more frequently in males, which may also reflect the influence of environmental and occupational exposures. These findings are consistent with previous reports indicating that both CLL and CML occur more commonly in men than in women [33,34].

Interestingly, our findings also revealed distinct age-related patterns in leukemia subtypes and associated chromosomal aberrations, reflecting both global trends and regional deviations. In the current study, ALL was most prevalent in children aged 1–10 years, aligning with global data identifying ALL as the most common pediatric cancer. Notably, among the ALL cases, two children were identified with Down syndrome, a known risk factor that increases ALL susceptibility up to 20-fold due to increased genetic susceptibility and dysregulated hematopoiesis [35]. Furthermore, in ALL cases, age-specific translocations such as t(12;21) and t(1;19) were observed predominantly in children under 12 years, consistent with previous studies identifying these abnormalities as common in pediatric ALL [36]. According to the literature, t(12;21) is associated with a favorable prognosis and improved survival with adequate treatment [37], whereas the t(1;19) translocation is linked to a poor prognosis and a higher risk of relapse, particularly in its unbalanced form [38]. In addition, high-risk aberrations such as t(9;22), the Philadelphia chromosome, were more common in patients under 12 years and in those aged 13–39 years among these cases. This aligns with its known higher prevalence and poorer prognosis in adult-onset ALL, driven by the *BCR::ABL1* fusion gene [39]. Similarly, patients with *11q23* aberrations exhibited a comparable age distribution, predominantly affecting those younger than 12 years and individuals aged 13–39. This finding is consistent with previous studies reporting that *11q23* (*KMT2A::MLL*) rearrangements occur in up to 80% of infant ALL cases, suggesting a likely prenatal origin during fetal hematopoiesis. These translocations are also associated with poor prognosis, with outcomes particularly unfavorable in younger infants [40].

In AML cases, the highest frequency was observed in young adults (21–40 years), diverging from global trends where AML predominates in older adults. This age distribution may reflect the high HIV burden in South Africa, which contributes to immune dysregulation and increased hematologic malignancy risk [41,42]. Among AML patients, chromosomal aberrations such as t(8;21) and t(15;17) were more commonly observed in younger individuals (13–39 years), aligning with existing literature that associates these abnormalities with early-onset AML subtypes [43]. The t(8;21) translocation is commonly associated with core-binding factor (CBF) AML, a genetically distinct subtype generally linked to a favorable prognosis [44,45]. Similarly, the t(15;17) translocation is a defining feature of acute promyelocytic leukemia (APL), a subtype of AML, although historically associated with early mortality, now carries a favorable prognosis due to its remarkable responsiveness to targeted therapies such as all-trans retinoic acid (ATRA) and arsenic trioxide [46]. Among these cases, the *11q23* chromosomal aberration was more frequently identified in patients younger than 12 and those aged 13–39 years. Translocations involving chromosome *11q23* are well-documented in pediatric AML and are often associated with poor prognosis [47]. Notably, t(9;22) chromosomal aberration was more frequently detected in AML cases aged 13–39 and 40–49 years. Although t(9;22) is rare in AML, previous studies have shown it tends to occur more often in adults, especially those over 40 years old, and it is generally associated with poor prognosis and resistance to conventional chemotherapy [48].

The CML cases were more frequently observed in middle-aged adults (41–60 years), aligning with data from African and Asian populations, where the median age at diagnosis was typically under 50 years [49]. The hallmark Philadelphia chromosome (t(9;22)) was observed across nearly all age groups except children among CML cases, reinforcing its role as a key diagnostic marker. In contrast, CLL was predominantly diagnosed in patients over 60 years, reflecting trends observed in Western populations [50]. These findings highlight both global consistencies and regional variations in leukemia subtypes [51,52,53], likely influenced by immune system aging (immunosenescence), which leads to a decline in immune surveillance, reducing the body’s ability to eliminate pre-leukemic cells and thereby increasing the risk of CLL development in older adults [54,55]. In CLL, deletions of *13q* and *17p* were the most frequent abnormalities, predominantly occurring in patients over 60 years. Importantly to note, isolated *13q* deletions are associated with early-stage disease, better response to treatment better outcomes (good prognosis) [22], while co-occurrence with *17p13.1* deletions is associated with poor prognosis and resistance to therapy [56]. The frequent co-occurrence of *13q34/13q14.3* with *17p13.1/11q22.3* observed in this study suggests a high-risk CLL subset potentially requiring more aggressive treatment approaches.

## 5. Limitations

This study’s findings are based exclusively on black participants, which may limit generalizability to other ethnic groups, as genetic backgrounds and chromosomal aberration patterns can vary across populations. Additionally, the study did not assess clinical outcomes, limiting the ability to correlate specific chromosomal abnormalities with prognosis, treatment response, or survival. The absence of HIV status data in the patient records represents a limitation of this study. Furthermore, the exclusion of probes such as del(11q), +12, and CCND1 may limit the comprehensive cytogenetic characterization of CLL.

## 6. Conclusions

This study provides critical insights into the prevalence and distribution of chromosomal aberrations in leukemia patients in a tertiary institution in Gauteng Province, addressing a significant gap in regional cytogenetic data. The findings confirm that CML was the most prevalent leukemia subtype in this population, aligning with trends observed in other sub-Saharan African studies. The current findings highlight the need for improved diagnostic and treatment capabilities tailored to local needs. Distinct chromosomal aberration patterns were identified across leukemia subtypes, such as the predominance of t(9;22) in CML, characteristic translocations in acute ALL and AML, and recurrent deletions in CLL, highlighting the unique genetic landscape and disease biology of this population. Gender and age-related differences further illustrate the complex interplay of genetic, hormonal, environmental, and the region’s high burden of HIV, and healthcare access factors influencing leukemia susceptibility and progression in African patients.

By characterizing these cytogenetic abnormalities, this study contributes foundational knowledge critical for developing targeted diagnostics, risk stratification, and personalized therapies tailored to African populations. Given the limited access to specialized leukemia care in many parts of Sub-Saharan Africa, expanding cytogenetic testing and molecular profiling is essential to improve early detection and personalized treatment outcomes.

## 7. Recommendation for Future Studies

Future research should investigate the genetic and environmental determinants of leukemia across diverse African populations, while healthcare systems work toward integrating advanced diagnostic technologies and tailored treatment protocols into routine care. Expanding cytogenetic and molecular testing capacity through wider access to techniques such as FISH, karyotyping, and next-generation sequencing in both tertiary and regional healthcare centers will be essential for early and accurate diagnosis. Additionally, multicenter and longitudinal studies across varied African regions are needed to map cytogenetic patterns, assess treatment outcomes, and develop robust prognostic models specifically suited to African populations. Collectively, these efforts will promote equity in leukemia care and contribute to reducing the disease burden across the continent.

## Figures and Tables

**Figure 1 diagnostics-15-02429-f001:**
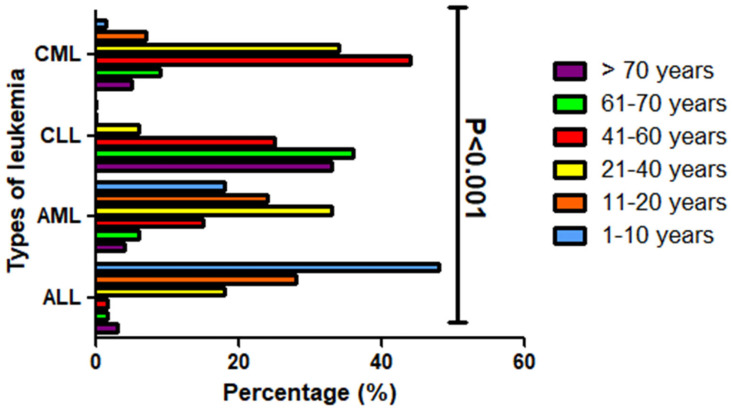
The distribution of various leukemia types among different age groups within the studied population. Each leukemia type is categorized based on its frequency within specific age ranges, visually represented by distinct color coding. The age groups are defined as follows: 1–10 years (blue); 11–20 years (orange); 21–40 years (yellow); 41–60 years (red); 61–70 years (green); >70 years (purple).

**Figure 2 diagnostics-15-02429-f002:**
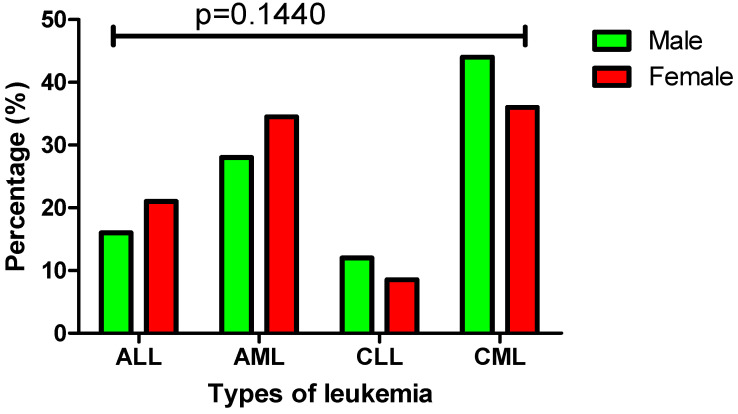
Comparison of leukemia types based on gender differences, the distribution of leukemia subtypes among male and female patients is visually represented using color coding—green for males and red for females.

**Figure 3 diagnostics-15-02429-f003:**
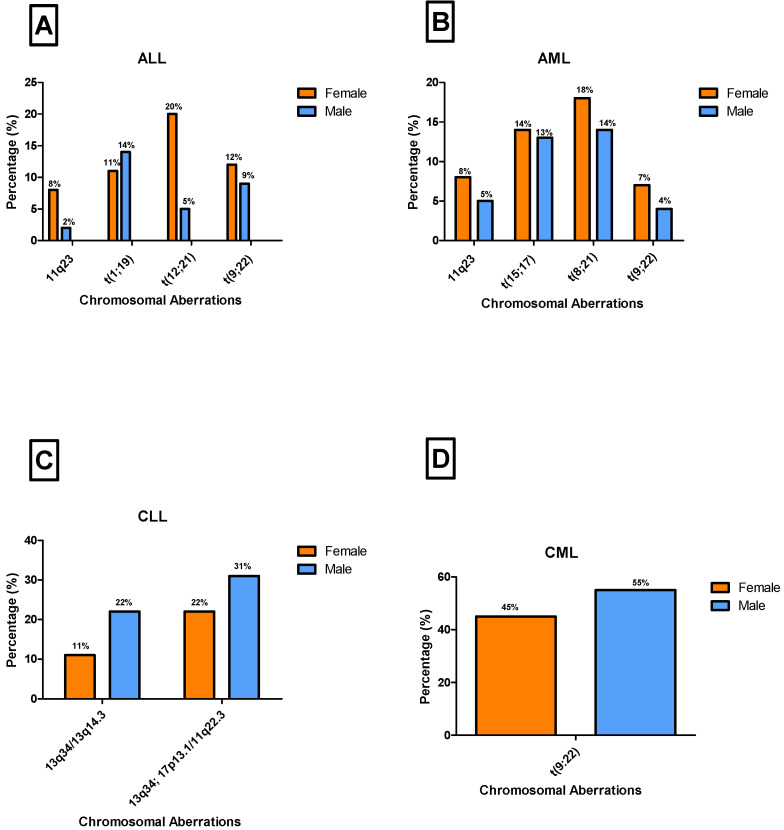
Gender comparison of highly prevalent chromosomal aberrations among leukemia patients in the study population. (**A**) Frequencies of chromosomal aberrations [*11q23*, t(1;19), t(12;21), and t(9;22)] in male and female patients with ALL. (**B**) Frequencies of chromosomal aberrations [t(8;21), t(15;17), inv(16), and t(9;11)] in male and female patients with AML. (**C**) Frequencies of chromosomal aberrations [*13q34/13q14.3* and *13q34/13q14.3*; *17p13.1/11q22.3*] in male and female patients with CLL. (**D**) Frequency of the chromosomal aberration [t(9;22)] in male and female patients with CML.

**Figure 4 diagnostics-15-02429-f004:**
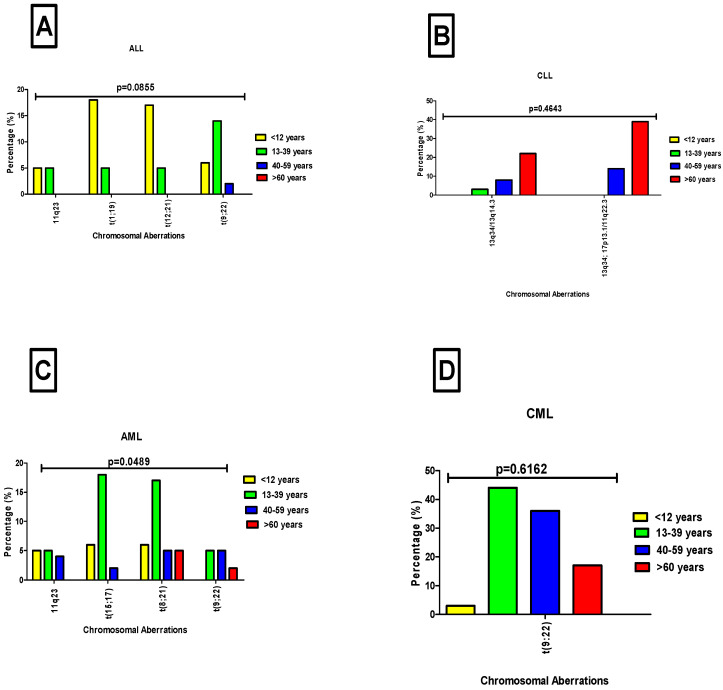
Distribution of chromosomal aberrations among leukemia patients across different age groups. Panels (**A,B**) present data for acute lymphoblastic leukemia and chronic lymphocytic leukemia cases, respectively. Panel (**C**) represents acute myeloid leukemia cases, while panel (**D**) represents chronic myeloid leukemia cases. Patients were grouped by age: <12 years (yellow), 13–39 years (green), 40–59 years (dark blue), and >60 years (red).

**Table 1 diagnostics-15-02429-t001:** The FISH Probe Panel used for the detection of chromosomal aberrations.

Leukemia Subtype	Probe	Target	Fluorophore Colour
CML, ALL, AML	*BCR::ABL1* Dual fusion	t(9;22)	Green (BCR), orange (ABL1)
AML	*RUNX1T1::RUNX1* (*ETO::AML1*)	t(8;21)	Orange (RUNX1T1), green (RUNX1)
AML	*PML::RARA*	t(15;17)	Orange (PML), green (RARA)
AML	*RARA* breaks apart	*RARA*	Orange/green (split signal for break apart)
AML	*CBFB* breaks apart	Inv(16:/t(16;16)	Orange/green (split signal for break apart)
ALL, AML	*KMT2A (MLL)*	*11q23*	Orange/green (split signal for break apart)
ALL	*ETV6::RUNX1 (TEL/AML1)*	t(12;21)	Orange (ETV6), green (RUNX1)
ALL	*TCF3::PBX1 (E2A::PBX1)*	t(1;19)	Orange (TCF3), green (PBX1)
CLL	*TP53* ATM	*17p13.1/11q22.3* deletion	Orange (TP53), green (ATM)
CLL	*D13S319::13q34* CEP12	*13q14.3* deletion/trisomy 12	Orange (D13S319), aqua (13q34), green (CEP12)

**Table 3 diagnostics-15-02429-t003:** Age distribution of leukemia subtypes in the study population (*n* = 349).

Age Groups	ALL (*n* = 65)	AML (*n* = 109)	CLL (*n* = 36)	CML (*n* = 139)
1–10 years (*n*, %)	31 (48%)	20 (18%)	0	2 (1.4%)
11–20 years (*n*, %)	18 (28%)	26 (24%)	0	10 (7%)
21–40 years (*n*, %)	12 (18%)	36 (33%)	2 (6%)	47 (34%)
41–60 years (*n*, %)	1 (1.5%)	16 (15%)	9 (25%)	61 (44%)
61–70 years (*n*, %)	1 (1.5%)	7 (6%)	13 (36%)	12 (9%)
>70 years (*n*, %)	2 (3%)	4 (4%)	12 (33%)	7 (5%)

**Table 4 diagnostics-15-02429-t004:** Patterns of the most prevalent chromosomal aberrations across acute and chronic leukemia subtypes in the study population.

Chromosomal Aberrations	Probe/Gene	ALL (*n* = 65), %	AML (*n* = 109), %	CLL (*n* = 36), %	CML (*n* = 139), %
t(1;19)	*TCF3::PBX1*	16 (25%)	0	0	0
t(12;21)	*ETO::RUNX1*	16 (25%)	0	0	0
t(8;21)	*RUNX1::RUNX1T1*	0	35 (33%)	0	0
t(15;17)	*PML::PARA*	0	29 (27%)	0	0
t(9;22)	*BCR::ABL1*	14 (22%)	12 (11%)	0	139 (100%)
*11q23*	*KMT2A*	6 (9%)	14 (13%)	0	0
*13q34/13q14.3*; *17p13.1/11q22.3*	*CEP12*; *TP53::ATM*	0	0	19 (53%)	0
*13q34/13q14.3*	*CEP12*	0	0	12 (33%)	0

## Data Availability

The original contributions presented in this study are included in the article. Further inquiries can be directed to the corresponding author.

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
