# Peer review of "Cytogenetic Profile of Chromosomal Aberrations in Leukemia Using the Fluorescence In Situ Hybridization (FISH) Method at a Tertiary Institution in Gauteng Province"

_diagnostics, 2025, doi:10.3390/diagnostics15192429_

Round 1

Reviewer 1 Report

Comments and Suggestions for Authors

The authors Duma Z et. al in the study “Cytogenetic Profile of Chromosomal Aberrations in Leukemia 2 using the Fluorescence in Situ Hybridization (FISH) Method at 3 a Tertiary Institution in Gauteng Province” have compared conventional karyotyping with Fluorescence In Situ Hybridization (FISH) in diagnosis utility of leukemia. The study deals with the conceptual flaw of comparison of two different techniques of diagnosis meant for different purpose. As in the abstract the authors give the statement of diagnostic efficacy of karyotyping is less than FISH in identifying chroromosomal translocation. This is a mis-leading statement as the karyotyping and FISH has different range of size in detecting chromosomal abnormalities and they cannot be replaced for each other.  For example, Karyotype can detect large chromosomal structural and numerical aberrations while FISH detects smaller deletions/duplication and other types of chromosomal abnormalities such as translocation etc.In addition the introduction as well as the result section describes leukemia and demographic details in general and not related to the topic itself.

Author Response

Dear Reviewer 1. 

We appreciate your time and effort in reviewing our manuscript. Kindly find attached the author's detailed responses to the reviewers' comments, provided below. 

Thank you.

Reviewer 2 Report

Comments and Suggestions for Authors

Cytogenetic profile of chromosomal aberrations in leukemia using the fluorescence in situ hybridization (FISH) method at a tertiary institution in Gauteng province

Zamathombeni Duma * , Karabo Charles Matsepane , Koketso Nkoana , Sara Mosima Pheeha , Bathabile Mbele , Tandekile Simela-Tshabalala , Donald Moshen Tanyanyiwa

Review comments:

  1. fluorescence in situ hybridization (FISH) method is widely used for help diagnosis and prognostication of hematologic (myeloid/lymphoid) malignancies in the USA and many other well-developed countries, which, however, might not be applicable in South Africa. Therefore, it is important for them to share their experiences in the field.

However, studying design is a problem. The articles can be shorten by focusing on comparing two methods (cyto vs FISH) in detection of acute and chronic leukemia. ?advantages and disadvantages.

  1. Table 1: it can be reorganized based on disease categories + probe set, easy to follow. Only test 13q and 17p seems not standard panel for CLL (del(11q), del(13q), +12, del(17p), and CCND1). Mantle call lymphoma should always be excluded from CLL. If only two probes are used for CLL, discussion for limitation is needed.
  2. Table 3. CML in younger age (<10 years), even <20 years, is uncommon. According to quick searching, the incidence of CML in patients <20 years old is only 2%. Did authors have confirmed the diagnosis according to CBC, blast count, morphology, immunophenotyping and PCR for p190 and p210 transcripts?
  3. Patients’ basic information is largely missing, and the results are difficult to follow. As not every CLL or ALL patient turns out to be FISH positive for del(13q) or del(17). It is unclear cases FISH positive for BCR::ABL1 : how many diagnosed B-ALL, or CML, or CML blastic phase.
  4. Paired studies are critical. What authors wanted to conclude and discuss herein is not quite clear. Figure and contents regarding age and gender distribution are too heavy and convey similar messages.

Author Response

Dear Reviewer 2,

We appreciate your time and effort in reviewing our manuscript. Please find attached the author's detailed responses to the reviewers' comments, provided below. 

Thank you.

Reviewer 3 Report

Comments and Suggestions for Authors

The manuscript by Duma et al. entitled: “Cytogenetic Profile of Chromosomal Aberrations in Leukemia using the Fluorescence in Situ Hybridization (FISH) Method at a Tertiary Institution in Gauteng Province” reports on 170 leukemia patients studied by using conventional cytogenetic and FISH. The study is aimed to compare the usefulness of the two methods to identify chromosomal abnormalities.

It is a retrospective study conducted on a limited number of cases and highlights findings mostly documented in literature.

Major revision:

  • In the Introduction the classification of leukemias suggested is inappropriate, the 5th edition of the WHO Classification of Haematolymphoid Tumors should be used or just refer that 4 acute and chronic hematological tumors have been included in the study
  • It is unclear whether patients included in the study are diagnosed consecutively or selected based on the outcome of chromosomal analysis/FISH
  • According to which criteria are FISH analyses performed?
  • If there are cases in which FISH was performed and found to be normal or failed, the proposed percentages should be revised.
  • In the 179 patients with no FISH or cytogenetic results, were the methods not performed or did they fail?
  • Are there cases with successful cytogenetics and failed FISH? If yes, it could be useful to have a table with the available data
  • Cases carrying recurrent translocations had additional abnormality?
  • Are the cases included in the study all primary acute leukemias? Are there any secondary leukemias?
  • How do the authors explain the frequencies of cases carrying the translocations t(15;17) and t(8;21) higher than those reported in Western countries?
  • Are there children with Down syndrome among ALL cases?
  • In CLL cases, please specify how many had 13q and 17p deletions and how many had complex rearrangement involving those regions.

Minor point:

Genes involved in fusions should be written in italics, separated by double colons.

The authors should clarify why were considered positive only cases in which a given genetic alteration was identified in 100% of the nuclei.

The discussion should be shortened as well as Sample Preparation Procedure and Laboratory Analysis

Author Response

Dear Reviewer 3, 

We appreciate your time and effort in reviewing our manuscript. Kindly find attached the author's detailed responses to the reviewers' comments, provided below.

Thank you.

Reviewer 4 Report

Comments and Suggestions for Authors

I strongly suggest that the authors improve the quality of their results, as much of their data lacks clarity and may confuse readers.

Author Response

Dear Reviewer 4,

We appreciate your time and effort in reviewing our manuscript. Please find attached the authors' detailed responses to the reviewers' comments, provided below for your review. 

Thank you.

Round 2

Reviewer 1 Report

Comments and Suggestions for Authors

The manuscript is improved after author's correction and can be accepted in the present form.

Reviewer 2 Report

Comments and Suggestions for Authors

Authors have answered questions appropriately and made significant improvement. The manuscript is ready for publication in the peer review journal.

Reviewer 3 Report

Comments and Suggestions for Authors

The Authors have complied with the suggestions proposed.

Reviewer 4 Report

Comments and Suggestions for Authors

No further comments at this stage.